# Circulating Irisin Levels in Patients with Chronic Plaque Psoriasis

**DOI:** 10.3390/biom12081096

**Published:** 2022-08-10

**Authors:** Francesca Ambrogio, Lorenzo Sanesi, Angela Oranger, Chiara Barlusconi, Manuela Dicarlo, Patrizia Pignataro, Roberta Zerlotin, Paolo Romita, Elvira Favoino, Gerardo Cazzato, Nicoletta Cassano, Gino Antonio Vena, Caterina Foti, Maria Grano

**Affiliations:** 1Department of Biomedical Science and Human Oncology, Unit of Dermatology, University of Bari, 70124 Bari, Italy; 2Department of Basic Medical Sciences, Neuroscience and Sense Organs, University of Bari, 70124 Bari, Italy; 3Department of Emergency and Organ Transplantation, Section of Human Anatomy and Histology, University of Bari, 70124 Bari, Italy; 4Department of Biomedical Science and Human Oncology (DIMO), University of Bari Medical School, 70124 Bari, Italy; 5Department of Emergency and Organ Transplantation, Pathology Section, University of Bari, 70124 Bari, Italy; 6Dermatology and Venereology Private Practice, 76121 Barletta, Italy; 7Dermatology and Venereology Private Practice, 70125 Bari, Italy

**Keywords:** irisin, psoriasis, Psoriasis Area and Severity Index

## Abstract

Irisin is an adipo-myokine, mainly synthetized in skeletal muscles and adipose tissues, that is involved in multiple processes. Only a few studies have evaluated serum irisin in psoriatic patients. This study aims to analyze serum irisin levels in patients with chronic plaque psoriasis, to compare them with values in controls, and to assess whether concentration of circulating irisin correlates with the severity of psoriasis, calculated by means of Psoriasis Area and Severity Index (PASI). We enrolled 46 patients with chronic plaque psoriasis; the control group included 46 sex- and age-matched subjects without any skin or systemic diseases. Serum irisin levels were measured by competitive enzyme linked immunosorbent assay. Our results showed a non-significant increase in serum irisin concentration in psoriatic patients compared to controls. A negative non-linear correlation between PASI and irisin levels was detected in psoriatic patients. Indeed, dividing patients according to psoriasis severity, the negative association between irisin and PASI was stronger in patients with mild psoriasis than in patients with higher PASI scores. Several control variables we tested showed no significant impact on serum irisin. However, erythrocyte sedimentation rate in the normal range was associated with significantly higher irisin levels in psoriatic patients. In conclusion, although irisin levels were not significantly different between controls and psoriatic patients, irisin was found to be negatively associated with psoriasis severity, especially in subjects with low PASI scores; however, further studies are needed to clarify the role of irisin in subjects with psoriasis.

## 1. Introduction

Psoriasis is a common chronic immune-mediated inflammatory disease with a prevalence ranging from 0.73% to 2.90% in Europe [1]. The most frequent form is chronic plaque psoriasis (psoriasis vulgaris), which accounts for about 70–80% of all variants and is characterized by well-demarcated erythematous plaques covered by silvery-white scales [2].

Recent evidence suggests that psoriasis should be regarded as a systemic inflammatory disease and not simply a disease limited to the skin [3,4]. Patients with psoriasis are at a higher risk of various comorbidities, such as Crohn’s disease, depression, nonalcoholic fatty liver disease, cardiovascular disorders, and metabolic syndrome or its components [3]. Up to 24% of patients with psoriasis have concomitant psoriatic arthritis (PsA) [5], which, in turn, exhibits a highly inflammatory systemic state [6]. The link between psoriasis and comorbidities such as metabolic syndrome and cardiovascular disease may be related to an underlying chronic inflammation with shared pathways and cytokines [7,8]. The association between psoriasis and obesity, defined on the basis of either body mass index (BMI) or abdominal fat, has been well established [8,9]. White adipose tissue is also an important source of hormones and bioactive mediators, such as adipokines, that can have a central role in linking the pathophysiological mechanisms of obesity and psoriasis [10].

Irisin is a 112 amino acid-adipomyokine produced by the proteolytic cleavage of the transmembrane glycoprotein named fibronectin type III domain-containing protein 5, as described in 2012 by Boström et al. [11]. Irisin was found to be secreted from skeletal muscle into the bloodstream in mice and humans during exercise through the expression of peroxisome proliferator-activated receptor-γ coactivator 1α [11]. Irisin is mainly synthetized in skeletal muscles and adipose tissues, although smaller amounts are secreted from other tissues [12,13].

After its identification, irisin was extensively studied and has been implicated in multiple processes, including white adipose tissue browning, regulation of energy expenditure, bone metabolism, and glucidic and lipidic homeostasis, and it was also shown to exert anti-inflammatory and neuroprotective effects [11,13,14,15,16]. Several studies assessed the involvement and concentration of irisin in different pathological conditions. For instance, lower levels of irisin have been detected in patients with coronary artery disease, type 2 diabetes mellitus and chronic renal disease [17,18,19], whereas circulating irisin was shown to be positively associated with insulin resistance and metabolic syndrome [20,21,22]. However, there is no conclusive evidence of such relationships yet. In addition, the link between irisin and BMI remains controversial [12,21]. The role of irisin in suppressing the risk factors of cardiovascular diseases has been suggested, although ambiguities still exist, and the underlying mechanisms have not been fully clarified [23,24,25].

Only a few studies have evaluated serum irisin in patients with psoriasis [26,27,28,29,30]. Indeed, the aim of our study was (i) to analyze serum irisin levels in patients with chronic plaque psoriasis, (ii) to compare them with values in control subjects, and (iii) to assess whether the concentration of circulating irisin correlates with the severity of psoriasis calculated by means of Psoriasis Area and Severity Index (PASI).

## 2. Materials and Methods

The study was approved by the Bioethical Committee of the University of Bari, Italy.

Eligibility criteria of cases were: age ≥ 18 years; diagnosis of chronic plaque psoriasis; signed informed consent; absence of history of malignancies; and no use of any systemic or topical drugs for treatment of psoriasis for at least one month prior to enrolment. For each patient the following data were recorded: demographic (age, sex) and anthropometric (height, weight, BMI) characteristics, PASI, psoriasis duration, history of PsA, and cardio-metabolic comorbidities (arterial hypertension, type 2 diabetes, dyslipidemia). In addition, we retrieved information about other comorbidities, which were included in a residual category.

The control group consisted of sex- and age-matched subjects who did not have any cutaneous or systemic acute or chronic diseases.

For all patients, blood samples were collected to determine serum levels of C-reactive protein (CRP), erythrocyte sedimentation rate (ESR) and irisin. Irisin serum concentrations were measured in cases and controls by competitive enzyme linked immunosorbent assay (ELISA; Cat. No. AG-45A-0046YEK-KI01, AdipoGen, Liestal, Switzerland), with a detection limit of 0.001 mcg/mL, intra-assay coefficient of variability (CV) of 6.9% and inter-assay CV of 9.07%. Notwithstanding challenges in the measurement of serum irisin levels [15,23], we adopted a comparable approach with respect to the recent literature [26,27,28,29,30].

All data were manually entered into an Excel spreadsheet and then analyzed by means of the R Statistical Software (version 4.0.3). Continuous variables were described through their mean, standard deviation (SD), median, interquartile range, minimum and maximum. Categorical variables were summarized by their shares in the sample. The Mann–Whitney test was used to detect serum irisin difference between psoriatic patients (cases) and controls.

In order to detect robust associations between irisin levels and PASI scores among psoriatic patients, we performed Spearman correlation and a series of ordinary least squares (OLS) regressions, accounting for a number of potential confounding factors (e.g., PsA, ESR, BMI, sex, age).

Taking into consideration the sample size of the study, *p*-values less than 0.05 were considered statistically significant. As a non-linear relationship between severity of psoriasis and irisin levels was suspected, the sample of psoriatic patients was divided into three groups depending on the severity defined by the PASI score. In particular, PASI values < 10 were classified as low (mild psoriasis), between 10 and 20 as moderate (moderate psoriasis) and >20 as high (severe psoriasis) [26].

ESR was regarded as a dichotomous variable, coded 1 if it was within the normal reference range and 0 if abnormal.

## 3. Results

Forty-six patients (31 males and 15 females) with chronic plaque psoriasis, recruited between September 2016 and February 2018 at the Dermatology Unit of University of Bari, in Southern Italy, participated in this study. None of the patients had previously received treatment with biologics for psoriasis and/or PsA. The mean age of patients was 52.5 years (SD 15.9), the average BMI was 27.0 kg/m^2^ (SD 6.25) and the mean PASI score was 13.3 (SD 9.57). Patients with mild psoriasis (39% of the sample) had an average PASI score of 4.66 ± 2.41, those with moderate psoriasis (37% of the sample) had a mean PASI value of 13.7 ± 3.15, whereas the mean PASI value in severe cases (24% of the sample) was 28.1 ± 5.11. The following comorbidities were recorded: PsA (26.3% of patients), arterial hypertension (17.4%), type 2 diabetes (6.5%), dyslipidemia (6.5%) and others (30.4%).

The control group was composed of 46 sex- and age-matched individuals without any cutaneous or systemic diseases.

Irisin levels in patients were marginally higher than in the control group (Figure 1). In particular, the median serum levels of irisin were 9.93 (interquartile range, 7.57–12.94) µg/mL and 9.32 (interquartile range, 6.97–12.19) µg/mL in cases and controls, respectively. However, the difference was not statistically significant (*p* = 0.201).

To assess the potential relationship between irisin and psoriasis severity, we performed Spearman correlation between irisin levels in all patients and their PASI scores. We found a negative correlation between these two variables (r= −0.308; *p* = 0.041) (Figure 2).

Comparing median irisin levels in each of the three groups (stratified on the basis of PASI scores) with levels in control subjects, we found that irisin was higher in patients with mild psoriasis compared to controls and to patients with more severe psoriasis (Figure 3). The Mann–Whitney test showed a significant increase in irisin in patients with low PASI scores versus controls (*p* = 0.01), and versus patients with moderate psoriasis (*p* = 0.04).

Moreover, we noticed that the intensity of the irisin-PASI relationship diminished, moving from the group with mild psoriasis to the group with severe psoriasis (Figure 4).

Finally, we performed a multivariate analysis to investigate which factors affect the relationship between irisin levels (in log) and PASI (Table 1). We estimated six models in order to control for different confounding factors and tested the robustness of the relationship between PASI and irisin levels. The use of a semi-log model allowed us to interpret the estimated coefficients as percentage variations. Among regressors we included PASI, its interaction with a dummy variable expressing the disease severity group (low, moderate and high PASI values) and a number of controlling factors. In addition, model 6 included squared PASI (PASI^2^) to account for a non-linear relationship.

There was no significant linear association between PASI and irisin levels (*p* = 0.455) in the full sample, while we found a negative yet non-linear correlation (see model 6 in Table 1, which points to a negative convex relationship).

Moreover, when we divided the sample into the three groups according to PASI values, the PASI score showed a negative and statistically significant association with irisin serum levels within each group, after controlling for BMI, age, sex (column 2 in Table 1) and duration of psoriasis (column 5). The robustness of such a negative relationship within the three groups was confirmed when ESR and PsA were included as additional controlling factors (columns 3 and 4 in Table 1). PsA, duration of psoriasis, BMI, sex and age showed no significant correlation with serum irisin levels.

Consistently with the results shown in Figure 4, the coefficients expressing the relationship between PASI and irisin levels increased when shifting from the low to high PASI groups. Indeed, our results suggested that an increase of 1 unit in the PASI score results in a decrease in irisin levels of approximately 12% in patients with mild psoriasis (−0.134 in our preferred specification), 6% in the intermediate group (−0.064) and 2% in the highest PASI group (−0.024).

In the multivariate analysis, we also investigated the influence of two reliable markers of systemic inflammation, ESR and CRP, on serum irisin. Our results indicated that irisin levels were 45% higher in psoriatic patients with an ESR score within the normal range compared with patients who had elevated ESR values (*p* = 0.036), while no significant relationship was detected for CRP (data not shown).

PsA did not significantly affect irisin levels (Table 1); similarly, other cardio-metabolic comorbidities (arterial hypertension, type 2 diabetes, dyslipidemia) had no impact on serum irisin (data not shown).

## 4. Discussion

Because of its involvement in the homeostasis of many systems, irisin has been recently at the core of a number of studies evaluating its direct and indirect roles in a variety of diseases. To the best of our knowledge, only five studies evaluated serum irisin in plaque psoriasis [26,27,28,29,30], showing conflicting results. In such studies, cases and controls were comparable in terms of age, sex and BMI, whereas, in our study, cases and controls were matched only for age and sex, as BMI data were unavailable for controls. Table 2 summarizes the main information regarding such studies and ours.

In particular, in previous studies, the comparison of irisin serum concentration between psoriasis patients and controls disclosed significantly lower values in cases in two studies [27,28], significantly higher values in cases in other two studies, one involving patients with mild, moderate and severe psoriasis [29], and the other recruiting patients with moderate-to-severe psoriasis [30]. Furthermore, Baran et al. reported higher values in psoriasis patients, though without statistical significance [26]. Our results indicated that the serum irisin concentration in psoriatic patients was marginally higher than in controls, though the size of such a difference was lower than in the study performed by Baran et al. [26]. This increase in irisin in the total sample of psoriasis patients could be related to patients with mild psoriasis, in whom the serum myokine concentration was significantly increased compared to the controls; however, in line with Baran et al. [26], the difference was not significant if we considered the total patients.

Previous research in psoriasis patients also demonstrated divergent findings concerning the relationship between serum irisin concentration and PASI scores, with no correlation shown in two studies [26,30], significant positive correlations found by other authors [28,29], and a significant negative correlation disclosed in another study [27].

Another finding emerging from our study concerns the relationship between irisin levels and the severity of psoriasis (as defined by PASI scores). Spearman’s correlation showed a significant negative correlation between irisin and PASI in patients with psoriasis, but this correlation was lost in the multivariate analysis. However, by dividing patients according to their psoriasis severity, we reported a heterogeneous significantly negative association within the PASI groups, indicating a non-linear relationship between the two parameters. Interestingly, our patients with higher PASI scores also had higher BMI values compared to patients with milder forms of psoriasis (data not shown), suggesting that metabolic alterations or mechanisms related to increased adipose tissue might contribute to determine irisin values in such a group of patients. Indeed, a negative and non-significant linear correlation was observed between serum irisin and BMI when considering the full sample of patients (Table 1), in line with previous results [26,30]. Accordingly, the role that irisin may play in overweight status or obesity is still controversial [12,17,21] and further studies are needed to shed light on the mechanisms that tie irisin to BMI.

Furthermore, we found that irisin in psoriatic patients did not significantly correlate with both the duration of the disease, in contrast to a previous study [26], and the presence of PsA. This contributes to a wider set of evidence about the association between irisin and both clinical conditions and laboratory markers.

Indeed, prior assessments of psoriasis patients showed no statistically significant correlations of serum irisin levels with glucose level [26,28,30], uric acid [30], insulin [28,30], homeostasis model assessment of insulin resistance (HOMA-IR) [27,28,30], lipid parameters [26,27,30], BMI [26,30] and metabolic syndrome and obesity [29], whereas Bulur et al. observed a significant negative correlation between serum irisin concentration and levels of serum triglycerides and LDL cholesterol and a positive correlation with serum HDL cholesterol [28].

In addition, Baran et al. noticed a significant positive correlation between irisin in psoriasis patients and laboratory parameters of inflammation, namely CRP and ESR, as well as lipocalin-2 [26], a protein secreted mainly by activated neutrophils that has been associated with inflammatory responses and described as elevated in psoriasis [31]. The results obtained by these authors suggested that irisin should not be considered an indicator of metabolic conditions, severity of psoriasis or efficacy of antipsoriatic treatment but might be a marker of inflammation in psoriatic patients [26]. In a similar way, in the study by Ozkok Akbulut et al. [30], irisin seemed to be associated with inflammation, as measured by high-sensitivity (hs)-CRP. The binary logistic regression analysis performed by the same authors demonstrated that CRP level and a BMI value > 30 kg/m^2^ were independent predictors of a higher irisin level. On the contrary, in the study carried out by Bulur et al. [28], there was no correlation between circulating irisin and hs-CRP.

Similarly, the results of a meta-analysis showed no significant correlation between circulating CRP and irisin levels, although a significant positive correlation was detected in overweight or obese subjects, and subgroup analyses revealed significant, positive, but weak correlations in cohort studies, studies involving healthy people, and studies with a male-to-female ratio < 1 [32].

Some studies have highlighted that irisin might have a role in immunity, as well as anti-inflammatory and anti-oxidative effects [32,33]. Moreover, the relationship between irisin and specific adipokines was previously investigated in psoriasis patients. In particular, Baran et al. found no correlation between irisin and the anti-inflammatory adiponectin and the pro-inflammatory leptin [26], Ozkok Akbulut et al. observed positive and strong correlations between irisin, adiponectin, and leptin [30], and Gamil et al. noticed a significant negative correlation between adiponectin and irisin [29]. Our results could not support the role of irisin as a reliable marker of inflammation in psoriasis patients. Indeed, in the current study, no significant relationship was found between irisin and CRP in patients with plaque psoriasis, while patients with ESR levels within normal range had significantly higher levels of serum irisin.

Overall, our findings are in contrast with some previous evidence in the literature. In general, inconsistencies might be related, at least partially, to the limited size of patient samples and different eligibility criteria of patients enrolled in such studies. It should be emphasized that the measurement of circulating irisin is still challenging because of the questionable accuracy of the available detection methods [15,23].

The main limitation of our study reflects a relatively small population involved in the analysis and the omission of further inflammatory stimuli in psoriatic patients that we cannot control for. For example, we collected anamnestic data expressing only the presence or absence of PsA, without a more precise quantification of its severity and activity. Furthermore, the presence of comorbidities or physiological conditions capable of interfering with irisin serum levels was not systematically assessed and was not generally regarded as an exclusion criterion for the recruitment of psoriasis patients in our study. Several factors, such as physical activity, diet, hormonal conditions, metabolic diseases and various other pathological disorders (e.g., renal failure, musculoskeletal and neurodegenerative diseases), as well as treatments for such conditions, may affect the circulating irisin levels [15,16].

However, our results are robust to a variety of control variables and model specifications. The stability of our quantitative estimates across different regression models (Table 1, columns from 2 to 5) suggests that our analysis successfully captured a robust and significant negative non-linear correlation between PASI and irisin levels.

## 5. Conclusions

The relationship between serum irisin concentration and psoriasis is still under-investigated. Previous literature shows inconclusive and often contrasting findings. Our study suggests a negative yet non-linear relationship between irisin and PASI scores. Indeed, we found robust evidence that, for a unitary increase in PASI scores, irisin levels diminished sensibly more in patients with mild psoriasis than in those with severe disease. To obtain this result, we ran several OLS regressions, controlling for possible confounding factors. Linear correlations within groups of patients with mild, moderate and severe psoriasis confirmed our results. Moreover, we found that irisin levels were marginally higher in psoriatic subjects compared with sex- and age-matched controls. Finally, we examined the correlation between irisin levels and two inflammatory markers (ESR and CRP), obtaining contrasting results, in partial opposition with some previous studies. Overall, our results suggest that irisin concentration in psoriatic patients displays a complex pattern, which still needs to be elucidated. Additional studies are necessary to understand the possible role of irisin in psoriasis and uncover the relationship between irisin levels and the severity of psoriasis.

Noteworthy, it would be welcome to examine the role of irisin within larger groups of patients with psoriasis and other inflammatory skin diseases in order to distinguish between those mechanisms that pertain just to psoriasis and those that have a more general validity.

## Figures and Tables

**Figure 1 biomolecules-12-01096-f001:**
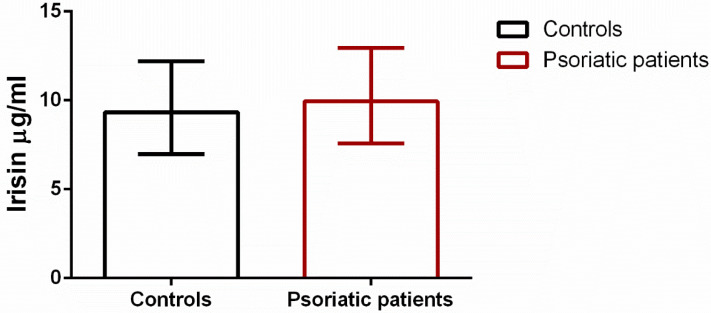
Comparison of median irisin levels in cases and controls. Whiskers indicate interquartile range.

**Figure 2 biomolecules-12-01096-f002:**
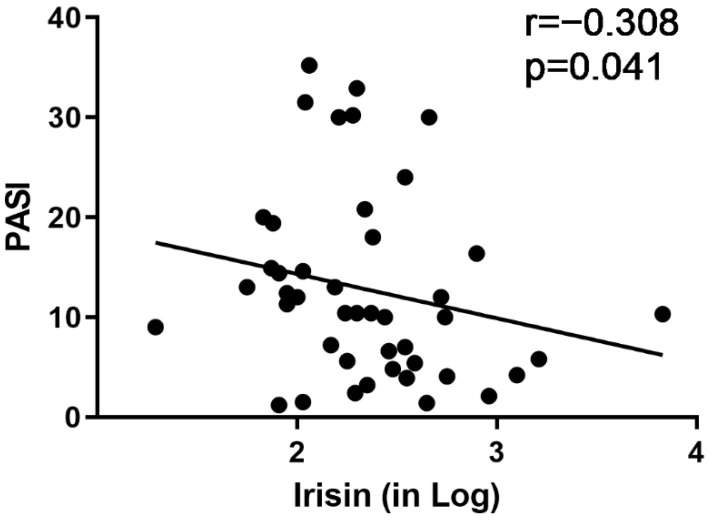
Irisin levels and PASI score for all cases. The logarithm of irisin levels has been used to better visualize the data and to reduce the skewness of their distribution.

**Figure 3 biomolecules-12-01096-f003:**
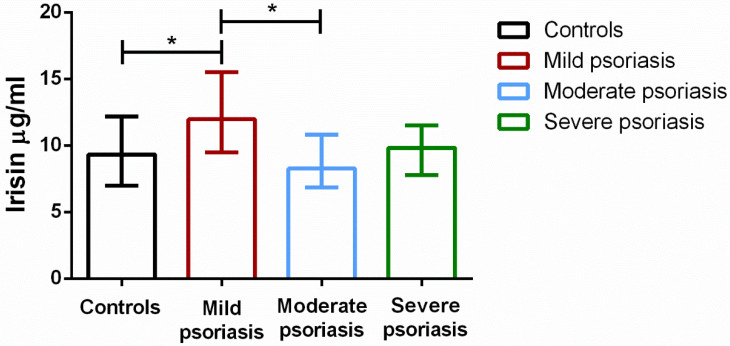
Irisin levels in groups of psoriasis patients stratified according to PASI score and controls. Whiskers indicate interquartile range. * *p* < 0.05.

**Figure 4 biomolecules-12-01096-f004:**
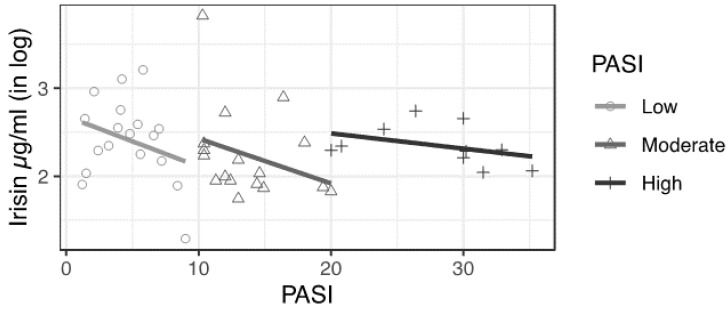
Relationship between irisin levels and PASI score among low, moderate and high PASI groups.

**Table 1 biomolecules-12-01096-t001:** Regression of irisin levels on PASI and other variables. Standard errors are reported in parentheses.

	Irisin (In Log)					
	(1)	(2)	(3)	(4)	(5)	(6)
PASI	−0.007					−0.085 *
	(0.009)					(0.034)
PsA			0.068			
			(0.175)			
ESR				0.457 *		
				(0.207)		
Duration of psoriasis					−0.008	
					(0.006)	
PASI^2^						0.002 *
						(0.001)
BMI	−0.014	−0.014	−0.016	−0.002	−0.012	−0.030
	(0.014)	(0.013)	(0.014)	(0.013)	(0.014)	(0.015)
Sex	−0.059	0.159	0.174	0.031	0.187	0.093
	(0.160)	(0.163)	(0.170)	(0.177)	(0.164)	(0.164)
Age	0.004	0.005	0.005	0.005	0.007	0.010
	(0.005)	(0.005)	(0.005)	(0.005)	(0.005)	(0.006)
PASI * group 1 (mild)		−0.118 *	−0.124 *	−0.134 **	−0.115 *	
		(0.046)	(0.048)	(0.046)	(0.046)	
PASI * group 2 (moderate)		−0.062 **	−0.064 **	−0.064 **	−0.060 **	
		(0.020)	(0.021)	(0.020)	(0.020)	
PASI * group 3 (severe)		−0.021 *	−0.021	−0.024 *	−0.021 *	
		(0.010)	(0.010)	(0.010)	(0.010)	
Constant	2.666 **	3.007 **	3.055 **	2.476 **	2.959 **	3.072 **
	(0.380)	(0.371)	(0.393)	(0.427)	(0.376)	(0.397)
Observations	39	39	38	36	38	39
R2	0.075	0.278	0.285	0.379	0.320	0.208
Adjusted R2	−0.034	0.142	0.118	0.223	0.161	0.088
Residual Std. Error	0.466 (df = 34)	0.424 (df = 32)	0.436 (df = 30)	0.410 (df = 28)	0.420 (df = 30)	0.437 (df = 33)
F Statistic	0.689 (df = 4; 34)	2.052 * (df = 6; 32)	1.710 (df = 7; 30)	2.437 ** (df = 7; 28)	2.017 * (df = 7; 30)	1.733 (df = 5; 33)

* *p* < 0.05; ** *p* < 0.01. BMI: body mass index; ESR: erythrocyte sedimentation rate; PASI: Psoriasis Area and Severity Index; PsA: psoriatic arthritis.

**Table 2 biomolecules-12-01096-t002:** Studies regarding irisin levels in adult patients with psoriasis vulgaris.

Study	Baran et al. [26]	Alatas et al. [27]	Bulur et al. [28]	Gamil et al. [29]	Ozkok Akbulut et al. [30]	Present study
Controls	15	30	37	50	43	46
Patients (mean age, yrs)	37 (48.6 ± 2.4)	30, receiving topical therapy (39.7 ± 9.3)	40 (38.0 ± 13.7)	50 (39.7 ± 13.1)	42 with PASI ≥ 10 (42.1 ± 11.6)	46 (52.5 ± 15.9)
PASI	median 18.8	mean 4.71 ± 3.52	mean 10.9 ± 5.6	mean 10.62 ± 5.84	mean 17.4 ± 7.8	mean 13.3 ± 9.57; median 11.3
Exclusion criteria	Systemic or topical treatment in the last 1 month, other types of psoriasis, chronic inflammatory or metabolic diseases interfering with psoriasis evaluation, dietary restriction	Other forms of psoriasis, systemic diseases *, pregnancy, breastfeeding, systemic treatment	Any systemic or topical treatment for psoriasis for the past 2 months, any treatment for any disease, PsA, other diseases **, pregnancy and lactation, alcohol and tobacco use	Any systemic treatment or ultraviolet therapy for psoriasis in the prior 6 months, systemic diseases	Systemic treatment (including phototherapy) in the last 3 months, pregnancy, BMI ≥ 35 kg/m^2^, diabetes, cancer	History of malignancies, systemic or topical treatment for psoriasis in the last 1 month (patients previously treated with biologics were not included)
ELISA for irisin measurement	BioVendor (catalog no. RAG018R)	No further details	BioVendor, Germany (catalog no. RAG018R)]	Sangon Biotech, China	Bioassay Technology Laboratory, China (catalog no. E3253Hu)	AdipoGen, Switzerland (catalog no. AG-45A-0046YEK-KI01)
Serum irisin levels in patients vs. controls	Higher but without statistically significant difference	Significantly lower	Significantly lower	Significantly higher	Significantly higher	Higher but without statistically significant difference
Correlation of irisin levels with PASI	Non-significant	Significant negative	Significant positive	Significant positive	Non-significant	Significant, negative non-linear

* Coronary artery disease, hepatic failure, renal failure, malignancy, etc. ** autoimmune or inflammatory diseases, diabetes mellitus, hypertension, metabolic syndrome, chronic infection, malignancy, thyroid, hepatic and renal diseases. BMI: body mass index; ELISA: enzyme-linked immunosorbent assay; PASI: Psoriasis Area and Severity Index; PsA: psoriatic arthritis.

## Data Availability

Not applicable.

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
