# Peer review of "Circulating Irisin Levels in Patients with Chronic Plaque Psoriasis"

_biomolecules, 2022, doi:10.3390/biom12081096_

Round 1

Reviewer 1 Report

The manuscript has been improved and further submission is possible.

Author Response

thank you very much

Reviewer 2 Report

Authors analyze serum irisin levels in patients with chronic plaque psoriasis, to compare them with values in controls, and to assess whether concentration of circulating irisin correlates with the severity of psoriasis, calculated by means of Psoriasis Area and Severity Index (PASI). They enrolled 46 patients with chronic plaque psoriasis; the control group included 46 sex- and age-matched subjects without any skin or systemic diseases. The results showed a non-significant increase of serum irisin concentration in psoriatic patients as compared to controls. A negative non-linear correlation between PASI and irisin levels was detected in psoriatic patients. They divide patients according to psoriasis severity, the negative association between irisin and PASI was stronger in patients with mild psoriasis than in patients with higher PASI scores. Several control variables they tested showed no significant impact on serum irisin. Only erythrocyte sedimentation rate in normal range was associated with significantly higher irisin levels in psoriatic patients. They concluded that, although irisin levels were not significantly different between controls and psoriatic patients, irisin was found to be negatively associated with psoriasis severity, especially in subjects with low PASI scores.

As the authors mention in Discussion section, reports on irisin have been inconsistent in their results. The reasons for those differences may be that sample sizes are small and ELISA kits are different for each of them. In this paper, though the sample size is not large, the authors have also conducted a detailed study of Irisin and each of its parameters. Then I have some questions.

major concerns)

1) What do you think of the curious correlation between PASI score and irisin in the overall mechanism by which psoriasis, adipocytes and skeletal muscle act? Combined with the results of other studies, can we conclude that irisin has no significant relationship with psoriasis itself?

2) Related to 1), why is the negative correlation so strong for mild psoriasis? How would you explain it? Is it conceivable that irisin is involved in the inflammation and early onset of psoriasis and that irisin may contribute to the severity of psoriasis? Does the fact that ESR within normal range correlates with irisin also suggest this possibility?

3) Since irisin is originally secreted predominantly by skeletal muscle, it is thought to correlate with the amount of skeletal muscle. In this study, you focus on BMI, which estimates the amount of body fat, but do not focus on the amount of skeletal muscle. A little research shows that there is a method called e-SMI, which uses calf circumlflex, body fat percentage, or body weight to estimate. Is there such an estimation method in the Italians (are all the subjects in this study Italian?)? If there is, it may be easier to correlate the irisin concentration with PASI score, if you correct it for skeletal muscle mass.

minor concerns)

1) line 75 cardiovascolar → cardiovascular

Author Response

This manuscript is a resubmission of an earlier submission. The following is a list of the peer review reports and author responses from that submission.

Round 1

Reviewer 1 Report

The study has been thoroughly designed and well written. The literature data are often conflicting and show that the relationship between serum irisin concentration and psoriasis is still underinvestigated. The authors suggest a negative yet non-linear relationship between irisin and PASI score but also they point that further on larger study groups studies are necessary to understand the possible role of irisin in psoriasis. I'd reccomend to emphasize more the conclusions in the abstract.

Reviewer 2 Report

1/ No type of paper is indicated at the beginning of the manuscript.

2/ Affiliations should be put superscripted after every name.

3/ Abstract does not include any conclusions.

4/ The subject that was investigated does not bring novelty, it has been already studied before several times in psoriatic patients and this paper is not going to increase our knowledge on this matter considering similarly low number of participants as in previous studies.

5/ Introduction section: provides the background for the study quite alright.

6/ Materials and methods section: the title and content of the template of this section was not deleted by the Authors and left in the submitted version of the manuscript...

The study involved relatively small number of participants, but the Authors correctly pointed it out in the limitations paragraph.

What does 'healthy controls' mean? I would avoid such expression. Do they just not have other dermatoses or they do not have any chronic diseases at all?

Apart from that, methods are quite thoroughly described.

7/ Results section: the tables have different font from the whole manuscript. Besides, quite thoroughly described.

8/ Discussion section: the headline is missing.

Indeed, the data on irisin in psoriatics are contradictory. This study compares everything with the previous studies and does not really bring novelty or allow to draw definite conclusions.

9/  Conclusions section: the conclusion is correct - studies on larger groups are needed because all the existing for now have been performed on small groups. This study does not increase our knowledge on this matter.